# Study on the Occurrence Difference of Functional Groups in Coals with Different Metamorphic Degrees

**DOI:** 10.3390/molecules28052264

**Published:** 2023-02-28

**Authors:** Jinzhang Jia, Yinghuan Xing, Bin Li, Dan Zhao, Yumo Wu, Yinuo Chen, Dongming Wang

**Affiliations:** 1College of Safety Science and Engineering, Liaoning Technical University, Fuxin 123000, China; 2Key Laboratory of Mine Power Disaster and Prevention of Ministry of Education, Liaoning Technical University, Huludao 125105, China; 3School of Environmental and Chemical Engineering, Shenyang Ligong University, Shenyang 110159, China; 4Faculty of Civil Engineering and Architecture, Zhanjiang University of Science and Technology, Zhanjiang 524000, China

**Keywords:** coal rank evolution, infrared characterization, structural parameters, functional group, fractional peak fitting

## Abstract

In order to quantitatively study the difference in occurrence content of functional groups in coals with different metamorphic degrees, the samples of long flame coal, coking coal, and anthracite of three different coal ranks were characterized by FTIR and the relative content of various functional groups in different coal ranks was obtained. The semi-quantitative structural parameters were calculated, and the evolution law of the chemical structure of the coal body was given. The results show that with the increase in the metamorphic degree, the substitution degree of hydrogen atoms on the benzene ring in the aromatic group increases with the increase in the vitrinite reflectance. With the increase in coal rank, the content of phenolic hydroxyl, carboxyl, carbonyl, and other active oxygen-containing groups gradually decreased, and the content of ether bonds gradually increased. Methyl content increased rapidly first and then increased slowly, methylene content increased slowly first and then decreased rapidly, and methylene content decreased first and then increased. With the increase in vitrinite reflectance, the OH-π hydrogen bond gradually increases, the content of hydroxyl self-association hydrogen bond first increases and then decreases, the oxygen-hydrogen bond of hydroxyl ether gradually increases, and the ring hydrogen bond first significantly decreases and then slowly increases. The content of the OH-N hydrogen bond is in direct proportion to the content of nitrogen in coal molecules. It can be seen from the semi-quantitative structural parameters that with the increase in coal rank, the aromatic carbon ratio *f*_a_, aromatic degree *AR* and condensation degree *DOC* increase gradually. With the increase in coal rank, *A*(CH_2_)/*A*(CH_3_) first decreases and then increases, hydrocarbon generation potential ‘*A*’ first increases and then decreases, maturity ‘*C*’ first decreases rapidly and then decreases slowly, and factor *D* gradually decreases. This paper is valuable for analyzing the occurrence form of functional groups in different coal ranks and clarifying the evolution process of structure in China.

## 1. Introduction

Coal molecules have a layered amorphous structure with short-range order and long-range disorder and have the characteristics of heterogeneity and insolubility, which makes people’s understanding of coal stay in the primary stage [1]. The structure of the coal body is closely related to the adsorption and spontaneous combustion of coal, and a correct understanding of the structure of the coal body can help to prevent natural coal and reduce the risk of gas disasters. FTIR technology is widely used in coal functional group analysis because of its advantages of short time consumption and no limitation of crystal structure [2].

Li Xia [3] carried out FTIR characterization of 28 coal samples with vitrinite reflectance ranging from 0.3% to 2.05%. The study shows that the structural evolution of coal is divided into three stages. There were two times of coalification jump correlation. Jia Tinggui [4] used FTIR technology to characterize five coal samples. His research showed that the distribution of functional groups in different coal ranks was different, but the evolution trend was similar. The active high functional groups gradually fell off, and the more stable functional groups gradually increased. Hao Panyun [5] carried out FTIR characterization of three kinds of coal-rank coal, and his research showed that coalification is the evolution process of carbon enrichment, dehydrogenation, deoxygenation, and removal of heteroatoms. Zheng Qingrong [6] used FTIR to characterize coal with moderate metamorphism and believed that the competition between aliphatic hydrocarbons and aromatics led to the transition of coalification in the fat coal stage. Yu Haiyang [7] believed that the activation energy of heteroatom groups, aliphatic hydrocarbons, and aromatic structures increased sequentially. Ibarra [8] used FTIR to characterize samples from peat to semi-anthracite and gave the assigned peak positions of each functional group. And it is proposed that the degree of aromatization increases with the degree of coalification, and the relative abundance of aromatic carbon to carboxyl groups is used as a parameter to evaluate coal maturity. Chen [9] obtained the Fourier transform infrared spectrum of peat to anthracite by FTIR and gave the chemical structure characteristics of three micro groups of vitrinite, chitinite, and inertinite. He [10] used FTIR and Raman spectroscopy to comprehensively characterize the chemical structure from lignite to anthracite and divided the coal structure evolution into three stages, thinking that with the increase in coal rank, aromatic clusters and graphite crystallite size Continuing to increase, the microcrystalline structure of coal is gradually improved and transformed into a graphite structure. Painter [11] used infrared spectroscopy to reveal the arrangement of hydrogen bonds, outlined the relationship between hydrogen bond strength and infrared frequency shift, and discussed the significance of hydrogen bonds in swelling measurements and solvent extraction.

Previous studies have revealed the fugitive states of functional groups in coal and studied the evolution of coal in detail, but relatively few previous studies have investigated the variation of the fugitive content of functional groups in different chemical fractions in different coal ranks. Therefore, in this paper, the Fourier infrared characterization technique is used to deconvolute the measured spectra by using Peakfit4.12 data processing software, and the long-flame, coking, and anthracite coals of the low, medium, and high-rank coals are characterized, and the functional group contents of each chemical component in the coals are quantified, and the relative contents and trends of each functional group in different coal rank coals are obtained, and the semi-quantitative structural parameters are calculated accordingly. Combining the information of functional group content and semi-quantitative structural parameters and giving the evolution law of each chemical component in different coal ranks, the evolution process of coal rank is discussed, and the structural evolution of coalification in different coal ranks and its mechanism are obtained.

## 2. Results and Discussion

### 2.1. Infrared Spectrum of Coal Sample

As shown in Figure 1 for FTIR spectra of different coal ranks.It can be seen from Figure 1a that the aromatic hydrocarbon structure is composed of three spectral peaks. With the increase in coal rank, the absorption peaks of isolated aromatic hydrogen and two adjacent hydrogen atoms per ring continue to strengthen, and each absorption peak moves to the direction of lower wavenumber tendency. In the coking coal and anthracite stages, a broad shoulder peak appears around 830 cm^−1^, and the absorption peak at 750 cm^−1^ representing the four adjacent aromatic hydrogen atoms gradually increases with coalification, reaches the maximum value in the coking coal stage, and then increases with the coal rank and decreased, which is consistent with the literature [12]. It can be seen from Figure 1b that with the increase in the degree of metamorphism, the absorbance of oxygen-containing functional groups decreases, especially around 1200 cm^−1^, and the absorbance gradually disappears, which proves that the evolution of coal is essentially a process of carbon enrichment and deoxygenation. The long-flame coal has a sharp peak at 1699 cm^−1^ representing the carboxyl group, and the peak disappears as the coal rank increases; carboxyl groups are only present in low-rank long flame coals; Figure 1c shows that with the increase in coal rank, the infrared spectral absorbance of aliphatic structure gradually decreased. With the increase in coal rank, The shoulder peaks around 2954 cm^−1^ and 2870 cm^−1^ broadened, the waveform became slower, and the methyl group content in this band increased. It can be seen from Figure 1d that from long-flame coal to anthracite, the absorbance of hydroxyl functional groups gradually decreases, and the shoulder peak of long-flame coal at 3220 cm^−1^ widens, indicating that the content of cyclic hydrogen bond increases. The peak positions of hydroxyl self-association OH groups absorption peaks gradually increased from 3408 cm^−1^ in long-flame coal to 3419 cm^−1^ in coking coal and finally to 3426 cm^−1^ in the anthracite stage. The peak height of coking coal at 3419 cm^−1^ is higher than that of coking coal and anthracite, and the content of self-associating hydrogen bonds is more abundant.

### 2.2. Aromatic Hydrocarbon Structure

In the FTIR spectrum of the coal sample, 900–700 cm^−1^ is the out-of-plane deformation vibration region of the aromatic structure CH, which is unstacked into 15–22 peaks, as shown in Figure 2. There are four kinds of aromatic ring substitutions: isolated aromatic hydrogen (900–850 cm^−1^), two adjacent aromatic hydrogen atoms per ring (850–810 cm^−1^), three adjacent aromatic hydrogen atoms (810–750 cm^−1^), four adjacent aromatic hydrogen atoms (750–730 cm^−1^) [13]. The four adjacent aromatic hydrogen atoms account for 26.54% of long-flame coal, 38.38% of coking coal, and 26.22% of anthracite. Its content first increases and then decreases, and its content increases because the aromatic ring is on the aromatic ring during the evolution from long-flame coal to coking coal. Oxygen-containing side chains and aliphatic side chains break off and fall off, resulting in an increase in the four adjacent aromatic hydrogen atoms. At the stage from coking coal to anthracite, the aromatic ring mainly undergoes condensation, resulting in a decrease in the four adjacent aromatic hydrogen atoms; The substitution mode of the benzene ring is dominated by three adjacent aromatic hydrogen atoms, and its absorption intensity represents the polycondensation direction of coal evolution, and the proportions of three adjacent aromatic hydrogen atoms are: 49.13%, 42.56%, and 40.08%, respectively. With the deepening of the degree of metamorphism, the substitution mode gradually decreased, indicating that coal molecules undergo condensation. With the increase in coal rank, the content of two adjacent aromatic hydrogen atoms per ring is 12.28%, 8.59%, and 12.73%, respectively, and the content first decreases and then increases. The isolated aromatic hydrogen decreased from 12.05% in long-flame coal to 10.49% in coking coal and finally increased to 21.97% in anthracite. The content of two adjacent aromatic hydrogen atoms per ring and isolated aromatic hydrogen evolution to the coking coal stage decreased, which is related to the shedding of oxygen-containing side chains and aliphatic side chains on the aromatic ring in this process, and then due to the condensation of aromatic rings, the content of two adjacent aromatic hydrogen atoms per ring and isolated aromatic hydrogen were both increased. The change in the substitution pattern of the aromatic nucleus is the result of various factors: substitution reaction of positioning groups on aromatic rings, dehydroaromatization of naphthenic, condensation of the aromatic ring, dehydroxylation, decarboxylation reaction on the aromatic ring, and aliphatic chain shedding [5]. The ratio (*DOS*) of the peak area around 870 cm^−1^ and the peak area around 750 cm^−1^ was chosen to reflect the degree of substitution of aromatic ring sites by functional groups [9]. Table 1 shows the relative percentages of the three integrated bands. With the increase in vitrinite reflectance, the *DOS* value generally showed a trend of increasing slowly at first and then increasing rapidly, indicating that the degree of substitution of hydrogen atoms on the aromatic ring increased. *A*_750_ cm^−1^ reaches the maximum in the coking coal stage, and the peak of 750 cm^−1^ infrared absorption peak of coking coal is the highest at this stage. During the transformation from low-rank coal to middle-rank coal, *A*_750_ cm^−1^ increases due to the oxygen-containing groups on the aromatic nucleus of the coal macromolecules fall off, and the aliphatic chain breaking off and falling off. *A*_870_ cm^−1^ increases due to naphthenic dehydroaromatization and condensation [14]. The conversion process from low-rank coal to middle-rank coal is mainly due to the shedding of oxygen-containing groups, and the *DOS* shows a slight upward trend. In high-mature anthracite coals, agglomerated aromatic structures predominate, with aromatic rings having an average size of 3 to 4 or more. The aromatization or condensation process can lead to increased concentrations of polynuclear aromatic structures in highly mature coals, resulting in aromatic nuclei. As the degree of substitution increases, the *DOS* value increases. With the increase in the degree of metamorphism, the four adjacent aromatic hydrogen atoms first increased and then decreased, the content of three adjacent aromatic hydrogen atoms gradually decreased, the content of two adjacent aromatic hydrogen atoms per ring and isolated aromatic hydrogen first decreased slightly and then increased gradually [15,16,17]. Three adjacent aromatic hydrogen atoms and four adjacent aromatic hydrogen atoms are the main ones. The degree of substitution of hydrogen atoms on the benzene ring in aromatics increases with the increase in vitrinite reflectance.

### 2.3. Oxygen-Containing Functional Groups

The absorption vibration region of the oxygen-containing functional group of the coal sample is located between [18] 1800–1000 cm^−1^ in the FTIR spectrum, and this interval is fitted to 22 peaks, as shown in Figure 3. In this interval, there are deformation vibrations of methyl and methylene groups, carbonyl, carboxyl, ether bonds, and C=C stretching vibrations in aromatic rings or fused rings. 1010 cm^−1^ belongs to the ash absorption vibration region, and its content is relatively low. 1036 cm^−1^ belongs to alkyl ether, and the content of alkyl ether in the process of coalification shows an upward trend, from 6.92% to 17.08%, and reaches 21.57% in the anthracite stage. 1097 cm^−1^ reflects the vibration of aryl ether. The aryl ether decreases from 5.44% to 5.25% and then slightly increases to 6.12%; The broad oblique band between 1330–1172 cm^−1^ is attributed to alcohol, phenol, C-O stretching vibration in ether. The content decreased from 28.91% to 9.42%, and finally decreased to 8.49%. The carbon-oxygen single bond in the alcohol hydroxyl group is longer than the chemical bond of other oxygen-containing functional groups, the chemical property is unstable, and it is prone to chemical reactions. In the process of coalification, the dehydroxylation reaction leads to the loss of oxygen and carbon elements. 1389 cm^−1^ belongs to the methyl symmetric stretching vibration, and its relative proportions in the coal are: 10.34%, 9.26%, and 11.46%, and its content first decreases and then increases; The antisymmetric deformation vibration of the methyl group and the scissor vibration of methylene group are reflected at 1442 cm^−1^, its content increased from 9.54% (long flame coal) to 16.61% (coking coal) and finally decreased to 10.65% (anthracite). The stretching vibration of aliphatic CHx increases first and then decreases, which is related to the enrichment of fatty substances during the first and second coalification jumps and the shedding of fatty substances during the third coalification process. The interval of 1610–1502 cm^−1^ is caused by C=C stretching vibration in aromatic or fused rings, and C=C increases from 28.25% to 32.86% and finally decreases to 29.88%. During the conversion from low-rank coal to medium-rank coal, the dehydroaromatization of naphthenes, epigenetic metamorphism, dynamic metamorphism, and pyrolysis effect lead to the shedding of side chain groups, and then the relative molecular weight of coal decreases, resulting in C=C content is relatively increased. The number of C=C in the agglomerated aromatic structure decreases with the increase in the size of the aromatic ring, so the condensation of the aromatic ring leads to the decrease in the C=C content in the high-maturity anthracite coal. The absorption peak at 1650 cm^−1^ is attributed to the conjugated carbonyl group, and C=O increases from 4.1% of the long-flame coal to 8.12% of the coking coal, which is related to the shedding of the oxygen-containing functional groups of the low-order long-flame coal, the exposure of active oxygen atoms, and the ether oxygen in the coal. Or the conversion of phenolic hydroxyl to C=O, and finally reduced to 5.32% of anthracite. 1696 cm^−1^ is an unsaturated carboxylic acid C=O. The carboxyl group has high reactivity. Its content is only 3.36% in long-flame coal and gradually disappears in coking coal and anthracite with the increase in coal rank. 1727 cm^−1^ is classified as aryl esters, and the content of aryl esters in coal samples is relatively low, with the content of 3.15%, 0.9%, and 1.47%, respectively. The ether bond is the main structure that forms the bridge bond connecting the basic structural unit of the condensed aromatic ring in the macromolecular structure of coal [4], and its chemical properties are stable. With the increase in coal rank, the content of active oxygen-containing groups such as phenolic hydroxyl group, carboxyl group, and carbonyl group gradually decreases, and the content of ether bond gradually increases; the amount of oxygen-containing functional groups are closely related to the generation of CO_2_ during coal evolution [12], during the evolution stage of low-rank coal to middle-rank coal, the rapid rupture of oxygen-containing functional groups leads to a large amount of CO_2_ gas generation, and carbon dioxide is mainly generated in the stage of *R_0_* < 1.3%. The essence of the coalification jump is the result of competition between aliphatic substances and aromatic substances, the increase in stable, functional groups, the decrease in unstable functional groups, the gradual decrease in oxygen-containing functional groups and aliphatic hydrocarbon content in coal, and the gradual increase in aromatics [19].

### 2.4. Structure of Aliphatic Hydrocarbons

The stretching vibration band of the aliphatic hydrocarbon structure in the FTIR spectrum of the coal sample is 3000–2800 cm^−1^, as shown in Figure 4, and this interval is fitted to six peaks. From the relative content of the three coal aliphatic hydrocarbons, the coal rank is from low to high. The absorption band near 2954 cm^−1^ belongs to the anti-symmetric methyl group, the content of the anti-symmetric methyl group is 5.81%, 7.28%, and 4.96%, respectively, and its content first increases and then decreases; 2924 cm^−1^ reflects the situation of the anti-symmetric methylene group, Anti-symmetric methylene decreased from 39.25% to 30.63%, and finally decreased to 13.8%. 2895 cm^−1^ is the stretching vibration of methine (-CH), and the proportion of methine is 28.32%, 17.41%, and 34.43%, respectively. The methine group is located in the aliphatic chain, and the connection between the saturated alicyclic ring and the aromatic ring and in the branch chain is connected with the methyl group [20]. The increase in its content indicates that there are a lot of branch chains in anthracite. 2879 cm^−1^ corresponds to the symmetrical methyl group, and the content of the symmetrical methyl group is 0%, 16.1%, and 20.58%, respectively. The symmetrical methyl group gradually increases with the degree of coalification. Near 2853 cm^−1^ belongs to symmetrical methylene, and the content of symmetrical methylene is 26.62%, 28.58%, and 26.23%, respectively. The total amount of methyl groups is 5.81%, 24.61%, and 25.54%, and the total amount of methylene groups is 55.85%, 59.21%, and 40.03%. On the whole, the methyl group content increases rapidly and then slowly increases, while the methylene content first slowly increases and then it decreases rapidly, and the methine content first decreases and then increases. In the coking coal stage, CH_2_ is in a strongly polar environment, surrounded by a large number of unsaturated structures that can provide electron pairs. Under the action of thermal metamorphism or dynamic metamorphism, only a small amount of energy can be provided to break [13].

### 2.5. Hydroxyl Structure

The 3650–3000 cm^−1^ in the FTIR spectrum of the coal sample is the stretching vibration band of the hydroxyl functional group in the coal, as shown in Figure 5, and the total unstacking is 6–8 sub-peaks. Hydroxyl is an important functional group that affects the reactivity of coal. It has a strong activation effect when breaking or forming cross-linking bonds and has a great impact on coal naturally [15]. Hydrogen bonds are used to connect the topologically distant and spatially compact parts of the network chain [11]. In this interval, hydroxyl groups form different hydrogen bonds with different hydrogen bond acceptors. The hydrogen bond force is 10 times that of the non-specific intermolecular force and plays an important role in the stability of the coal molecular network [13]. The sharp peak near 3616 cm^−1^ is the free hydroxyl stretching vibration region, This is because the hydrogen bond between the hydroxyl groups cannot be formed due to steric hindrance, or the hydrogen bond is very weak. Most scholars believe that this peak is the vibration of free hydroxyl groups in coal, and some scholars attribute it to the crystal water in clay minerals [16]. The contents of free hydroxyl groups are 2.96%, 5.99%, and 4.46%, respectively. Free hydroxyl groups have higher activity than alcohols and phenols and can react with other functional groups at low temperatures to convert into carbonyl or carboxyl groups. The occurrence content in coal is low. The hydrogen bond formed between the hydroxyl group and the π electron cloud on the aromatic ring is called the OH-π hydrogen bond, and the absorption peak is [21] located near 3540 cm^−1^; The OH-π hydrogen bond decreases from 8.9% of long flame coal to 4.32% of coking coal, and finally increases up to 17.6%, with the increase in coal rank, the degree of aromatization increases, and the density of π electron clouds on the aromatic ring increases, and the probability of combining with hydroxyl groups to generate OH-π hydrogen bonds increases. 3430 cm^−1^ is a self-associative hydrogen bond of the hydroxyl group, and its contents are: 31.35%, 51.72%, and 37.38%, and its contents increase first and then decrease; As the clusters fall off, coal molecules tend to have a compact three-dimensional structure, the spacing between aromatic sheets gradually decreases, and the probability of hydroxyl self-association hydrogen bond formation increases [5]. Anthracite is coal with high maturity, and the hydroxyl groups are gradually removed during the coalification process, resulting in a decrease in the content of self-associative hydrogen bonds generated in anthracite. 3330 cm^−1^ belongs to the hydroxyl ether hydrogen bond, OH-O hydrogen bond is the hydrogen bond formed by OH and ether oxygen, the relative contents are 22.21%, 24.91%, 25.52%. With the increase in coal rank, the content of ether bonds gradually increased, and the internal arrangement of the molecules tended to be closer, which led to an increase in the probability of hydrogen bonding between hydroxyl and ether oxygen. The 3220 cm^−1^ stretching vibrations are attributed to cyclic hydrogen bonds, and the content of cyclic hydrogen bonds decreased from 32.18% to 4.57% and finally increased to 9.23%; with the increase in vitrinite reflectance, coal molecules undergo dehydroxylation reaction, The arrangement of the aromatic lamellae gradually tends to be parallel, and the probability of the hydroxyl groups forming cyclic hydrogen bonds in space decreases. The OH-N hydrogen bond is an acid-base complex formed between phenol and pyridine. 3040 cm^−1^ belongs to the hydroxyl-nitrogen-hydrogen bond. The content of the OH-N hydrogen bond is 2.4%, 8.49%, and 5.81%, respectively. Its content corresponds to the content of nitrogen in the elemental analysis.

### 2.6. Calculation and Analysis of Semi-Quantitative Structural Parameters

The structural differences of coals of different ranks are difficult to be directly distinguished by the spectrum and need to be analyzed in combination with the relevant semi-quantitative structural parameters. The author selects the following semi-quantitative structural parameters, as shown in Table 2 to analyze the evolution process of the coal macromolecular structure. The calculation results of the relevant structural parameters are shown in Table 3.

The aromatic carbon ratio *f_a_* is the percentage of carbon atoms that characterize aromatic hydrocarbons. It is assumed that there are only two carbon atoms in coal, aromatic carbon and aliphatic carbon, and it is calculated by Equation (1).
(1)$$f_{a}=\frac{C_{\mathrm{ar}}}{C}=1-\frac{C_{\mathrm{al}}}{C}=1-\left(\frac{H_{\mathrm{al}}}{H}\times \frac{H}{C}\right)/\frac{H_{\mathrm{al}}}{C_{\mathrm{al}}}$$
(2)$$\frac{H_{\mathrm{al}}}{H}=\frac{A_{3000\sim 2800}}{A_{3000\sim 2800}+A_{900\sim 700}}$$
(3)$$\frac{H}{C}=\frac{H_{\mathrm{ad}}}{C_{\mathrm{ad}}/12}$$

In Equation (1), C_al_/C is the relative content of aliphatic carbon in total carbon; H_al_/H is the relative content of aliphatic hydrogen in total hydrogen, which can be obtained from Equation (2); H/C is the ratio of hydrogen to carbon atoms, which can be obtained from Equation (3); H_al_/C_al_ is the ratio of hydrogen to carbon atoms in aliphatic hydrocarbons, generally 1.8;

The aromaticity *AR* is expressed by the ratio of aromatic hydrocarbons to aliphatic hydrocarbons, which represents the degree of aromatization of coal [22]. The degree of condensation *DOC* is represented by the relative abundance of the out-of-plane deformation vibrations of aromatic CH at 900–700 cm^−1^ and the C=C stretching vibrations of aromatic rings at 1600 cm^−1^ [23]. *A*(CH_2_)/*A*(CH_3_) represents the length of aliphatic chains and the degree of branching [19]. The hydrocarbon generation potential ‘*A*’ uses the aliphatic hydrocarbons at 3000–2800 cm^−1^. For the change of C=C content of 1600 cm^−1^, the larger the ‘*A*’ factor value, the greater the hydrocarbon generation potential of the coal. The smaller the ‘*A*,’ the greater the maturity [20,24]. The maturation level ‘*C*’ of coal is represented by the change of C=O to C=C stretching vibration. The lower the value, the greater the maturity. The *D* parameter is represented by the relative abundance of C=O bonds and C-O content, which can reveal the evolution law of different oxygen-containing functional groups.

#### 2.6.1. Evolution of Aromatic Structure

It can be seen from Table 3 that with the increase in coal rank, the aromatic carbon rate *f_a_* increases from 0.755 (long flame coal) to 0.863 (coking coal) and finally increases to 0.898 (anthracite), the proportion of aromatic carbon increases continuously, and the aromatic degree *AR* increases from 0.44 for long flame coal to 1.13 for coking coal and finally increases to 3.29 for the anthracite stage. As the degree of coal metamorphism increases, the degree of aromatization of coal continues to increase. There are three reasons for this phenomenon. First, the break-off of aliphatic chains and oxygen-containing functional groups results in a decrease in the molecular weight of coal and a relative increase in the content of aromatic hydrocarbons; Secondly, the deoxygenation and degreasing reactions generate a large number of active sites, which are cross-linked with each other, and the polycondensation of aromatic rings leads to the enlargement of the aromatic condensed system; and the dehydroaromatization of alicyclic rings. Whether the increase in aromatic carbon rate is caused by the shedding of aliphatic hydrocarbons and oxygen-containing side chains or by the condensation of aromatic rings needs to be further analyzed in conjunction with the degree of condensation: As the coal rank increases, the degree of condensation *DOC* gradually increases, and the increase in the degree of condensation of the aromatic nucleus is due to the breakage of the aliphatic side chain on the aromatic ring, the breakage of the oxygen-containing functional group, and the generation of a large number of active sites, which intersect with each other. Linked, agglomerated aromatic rings undergo polycondensation. During the evolution of molecular structure from low-rank long flame coal to medium-rank coking coal, the oxygen-containing functional group breaking off and aliphatic ring dehydrogenation aromatization are dominant, while during the transformation from medium-rank coking coal to high-rank anthracite, the degree of aromatic ring crosslinking increases, and aromatic ring condensation plays a major role [22,23].

#### 2.6.2. Evolution of Aliphatic Structure

With the increase in vitrinite reflectance, *A*(CH_2_)/*A*(CH_3_) first decreased and then increased. There are two reasons for the decrease in *A*(CH_2_)/*A*(CH_3_) from the long flame coal to the coking coal stage: On the one hand, the aliphatic hydrocarbon breaks and falls off, causing the methylene content to decrease; on the other hand, the degree of branching increases caused by an increase in methyl groups, alicyclic action leads to a decrease in methylene groups [12]. The ‘*A*’ factor of hydrocarbon generation potential increases at this stage, indicating that the content of aliphatic hydrocarbons relative to aromatic hydrocarbons increases at this stage, so the degree of aliphatic branching increases, and the alicyclic effect leads to *A*(CH_2_)/*A*(CH_3_) decline. From coking coal to anthracite, a large number of aliphatic hydrocarbons are broken and removed, the alicyclic ring is continuously broken, and the methylene functional group increases, resulting in the increase in *A*(CH_2_)/*A*(CH_3_). Of course, the increase in *A*(CH_2_)/*A*(CH_3_) may also be caused by a large amount of methyl shedding. In the aliphatic hydrocarbon system, the stability of methyl is higher than that of methylene, so it is not the result of methyl shedding. With the increase in coal rank, the degree of branching first increased and then decreased, and the length of the aliphatic chain decreased first and then increased. The increase in methine content reflects an increase in the degree of branching. The hydrocarbon generation potential ‘*A*’ first increases and then decreases with the vitrinite reflection. It reaches the maximum in coking coal, which means that from low-rank long-flame coal to the coking coal stage, coalification is mainly the enrichment of aliphatic substances in this stage. At the stage of anthracite with high maturity, aliphatic substances undergo pyrolysis, break and fall off, resulting in active sites, and some aliphatic chains undergo aromatization to form aromatic groups. The increase in aromatic systems is the main process. Compared with low-rank coal, high-rank anthracite has generated a large amount of hydrocarbon gas under the action of coalification, and the hydrocarbon generation potential of high-rank anthracite is lower than that of low and medium-rank coal [24,25,26]. The decrease inen ‘*A*’ factor is related to the second coalification jump [9].

#### 2.6.3. Evolution of Oxygen-Containing Functional Groups

With the increase in coal rank, the content of oxygen-containing functional groups gradually decreased, and the maturity of ‘*C*’ first decreased rapidly and then decreased slowly. The lower the value, the greater the maturity. During the conversion of low-rank long-flame coal to medium-rank coking coal, oxygen-containing groups such as aryl esters, carbonyl groups, and carboxyl groups are mainly shed. Forming active sites causing further condensation of aromatic rings, resulting in an increase in C=C stretching vibrations in aromatic or fused rings, resulting in a rapid decrease in maturity ‘*C*.’ From middle-rank coking coal to high-maturity anthracite coal, due to the massive removal of aliphatic substances and oxygen-containing groups at low and medium coal ranks, the shedding rate of a series of oxygen-containing functional groups such as C=O slows down. There are many active sites, and these active sites are cross-linked with each other, and the agglomerated aromatic rings undergo polycondensation and aromatization, resulting in the enlargement of the aromatic system. Therefore, in the process of converting from middle-rank to high-rank coal, the increase in C=C content leads to the decrease in the ‘*C*’ factor. The *D* parameter can reveal the evolution of different oxygen-containing functional groups. With the increase in coal rank, the *D* factor first decreases rapidly and then decreases slowly. The C-O content in ether and alkyl ether increased gradually. In the process of converting from long-flame coal to coking coal, the rapid decrease in the *D* parameter is mainly because, at this stage, the carboxyl groups, carbonyl groups, and aromatic lipids with higher activity are rapidly removed, and some C=O is broken with aliphatic hydrocarbons and aromatic rings. The active sites recombine to form less reactive alkyl and aryl ethers [27,28]. From the coking coal to the anthracite stage, the decreasing range of the *D* factor decreases, and the shedding rate of C=O is still greater than the decreasing rate of C-O content at this stage.

## 3. Experimental Part

### 3.1. Coal Quality Analysis

The anthracite coal of Shanxi Yangmei No.2 Mine, the coking coal of Shaqu No.1 Mine, and Qinggangping long-flame coal were selected as the research objects. The coal samples are prepared according to the provisions of the preparation method of coal samples (GB474-2008). The coal samples in the air-dry base state are put into the crusher, and vibrating screen for repeated crushing, screening, and shrinkage, and the samples with a particle size of more than 200 mesh are made for sealing and preservation [29]. The 5E-MAG6600 industrial analyzer was used to measure the moisture, ash, and volatile content of coal samples. The content of C, H, N, and S in coal samples was determined by Vario EL elemental analyzer. The industrial analysis and elemental analysis results are shown in Table 4.

### 3.2. Coal Sample Preparation and Test Conditions

The FTIR test was performed using a Thermo Scientific Nicolet IS5 Fourier transform infrared spectrometer. The instrument has a spectral range of 4000~400 cm^−1^ and a resolution of 0.04 cm^−1^. Take 10 g of 400 mesh coal samples and mix with KBr in a mass ratio of 1:200, and grind them to make transparent pastille with a thickness of 0.2–0.5 mm. The pressed pastille is placed in an oven at 110 °C to dry for 4 h and then taken out. Placed on the sample holder for measurement, the number of scans was 32 times, and the interference of water was deducted during scanning [12].

### 3.3. Peak Fitting Software and Method

The occurrence state of each element in coal is complex, and there are many functional groups of various groups. The measured spectrum is the result of the superposition of various spectral peaks, and it is difficult to obtain information directly. Therefore, this study uses the Peakfit4.12 data processing software to solve the measured spectrum. Overlap processing uses the baseline calibration method based on linearity or compensation for the selected area, determines the position and number of the initial unstacked fitting peaks according to the second derivative of each spectral line, and preset the peak shape of the component peaks as Gaussian and Lorentzian It is also reasonable to select all peaks as Gaussian peaks in the fitting process [29]. The fitting criterion is that the residual sum of squares between the original spectral line and the fitted spectral line is the minimum objective function.

The FTIR spectra of coal samples with different degrees of metamorphism are shown in Figure 6. The FTIR spectra of coal samples can be divided into four intervals according to the absorption peaks, which are hydroxyl functional groups between 3650–3000 cm^−1^, aliphatic hydrocarbon groups between 3000–2800 cm^−1^, Oxygen-containing functional groups between 1800–1000 cm^−1^, aromatic structures between 900–700 cm^−1^ [30].

## 4. Conclusions

(1)With the increase in the degree of metamorphism, the four adjacent aromatic hydrogen atoms first increased and then decreased, the content of three adjacent aromatic hydrogen atoms gradually decreased, the content of two adjacent aromatic hydrogen atoms per ring and isolated aromatic hydrogen decreased slightly at first and then increased gradually. Three adjacent aromatic hydrogen atoms and four adjacent aromatic hydrogen atoms are dominant, and the degree of substitution of hydrogen atoms on the benzene ring in aromatics increases with the increase in vitrinite reflectance. The content of active oxygen-containing groups such as phenolic hydroxyl group, carboxyl group, and carbonyl group gradually decreased, and the content of ether bond gradually increased.(2)With the increase in vitrinite reflectance, the methyl content first increased rapidly and then slowly increased, the methylene content first slowly increased and then decreased rapidly, and the methine content first decreased and then increased. OH-π hydrogen bonds gradually increased, self-associated OH groups bonds were the main type of hydroxyl hydrogen bonds in coal molecules, and their content first increased and then decreased, the OH-ether hydrogen bonds gradually increased, and the cyclic hydrogen bonds first decreased significantly and then slowly The OH-N hydrogen bond content is proportional to the nitrogen content in the coal molecule.(3)With the increase in coal rank, the aromatic carbon rate *f_a_* and the aromatic degree *AR* gradually increase. During the evolution of the molecular structure from low-rank long-flame coal to middle-rank coking coal, oxygen-containing functional groups break off and alicyclic dehydrogenation. aromatization is dominant. During the evolution from middle-rank coking coal to high-rank anthracite coal, the degree of cross-linking of aromatic rings increases and aromatic ring condensation is dominant. *A*(CH_2_)/*A*(CH_3_) decreased first and then increased. During the evolution, the degree of branching of aliphatic substances increased, and alicyclization and the increase in methyl content led to the decrease in *A*(CH_2_)/*A*(CH_3_); From coking coal to anthracite, a large number of aliphatic hydrocarbons are broken and removed, the alicyclic ring is continuously broken, and the methylene functional group increases, so that *A*(CH_2_)/*A*(CH_3_) increases. The hydrocarbon generation potential ‘*A*’ first increased and then decreased mainly because: from the long flame coal with lower coal rank to the coking coal stage, the coalification is mainly the enrichment of aliphatic substances in this stage. At the stage of anthracite with a higher maturity, the aliphatic substances undergo pyrolytic fracture and fall off.(4)With the increase in coal rank, the content of oxygen-containing functional groups gradually decreased, and the maturity ‘*C*’ decreased rapidly at first and then decreased slowly. The main reason is that in the process of converting from long-flame coal to coking coal, carbonyl, carboxyl, aryl esters, and other highly active oxygen-containing functional groups fall off, and C=C in aromatic rings or fused rings are formed as the main process. During the evolution process from coking coal to anthracite, the shedding rate of C=O slowed down, and the condensation and aromatization of agglomerated aromatic rings occurred, and the ‘*C*’ factor decreased slowly at this stage. The *D* factor gradually decreased, mainly because the more active carboxyl groups, carbonyl groups, and aryl esters were rapidly removed, and some carbon-oxygen double bonds were broken and recombined with the active sites on aliphatic hydrocarbons and aromatic rings to form less active alkyl ethers and aryl ether.

## Figures and Tables

**Figure 1 molecules-28-02264-f001:**
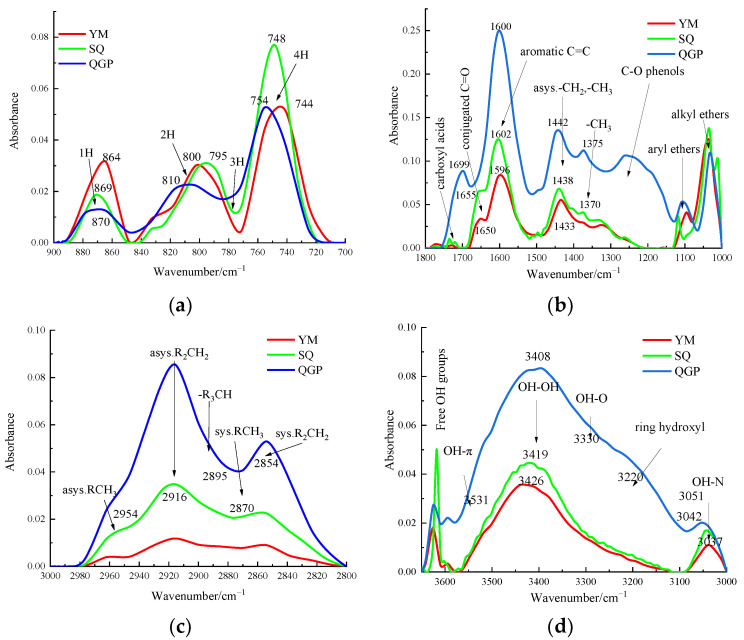
FTIR spectra of different coal ranks. (**a**) Aromatic hydrocarbon; (**b**) Oxygen-containing functional group; (**c**) Aliphatic structure; (**d**) Hydroxyl functional group.

**Figure 2 molecules-28-02264-f002:**
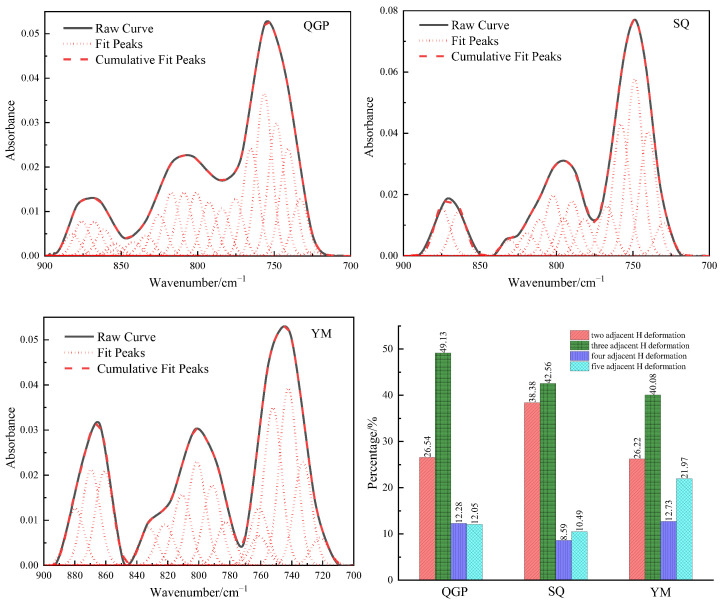
Fitting results and occurrence content of Aromatic hydrocarbon in coal samples.

**Figure 3 molecules-28-02264-f003:**
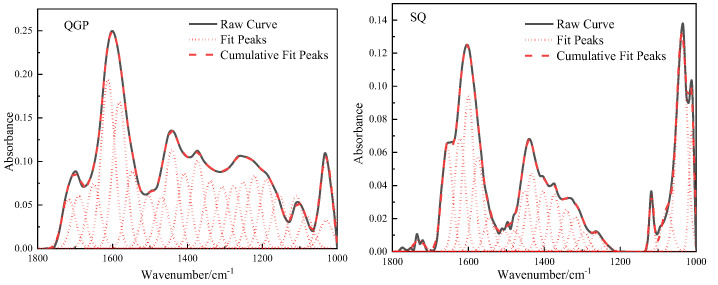
Fitting results and occurrence content of oxygen-containing functional groups in coal samples.

**Figure 4 molecules-28-02264-f004:**
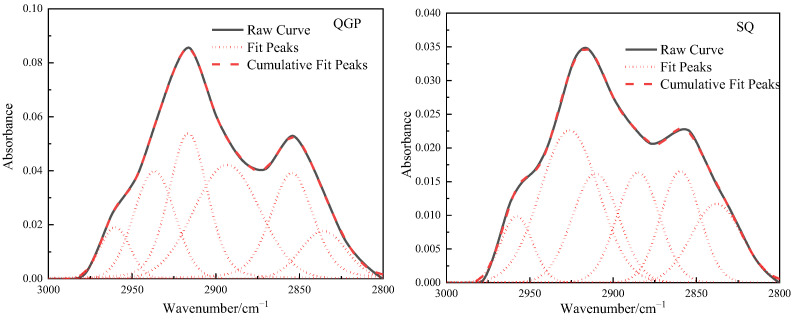
Fitting results and occurrence content of aliphatic hydrocarbons in coal samples.

**Figure 5 molecules-28-02264-f005:**
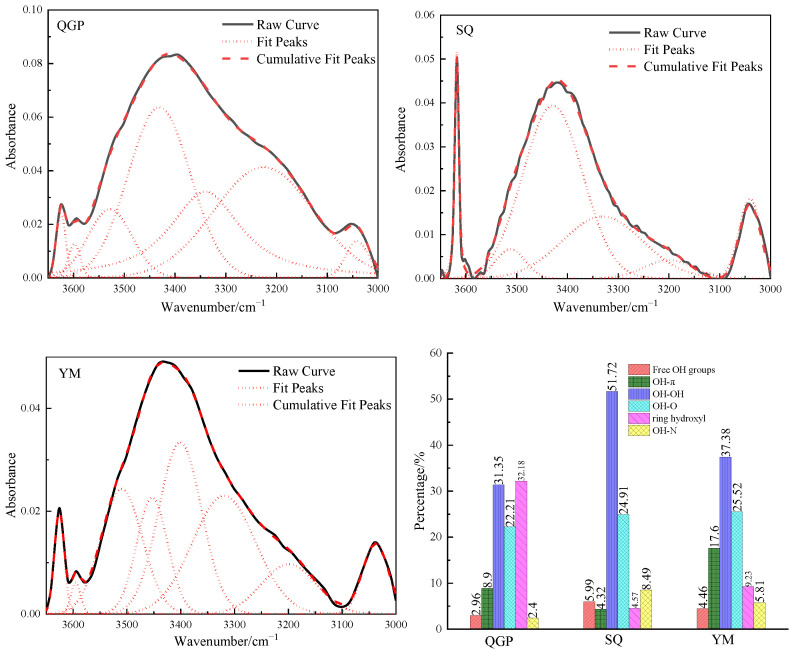
Fitting results and occurrence content of hydroxyl in coal samples.

**Figure 6 molecules-28-02264-f006:**
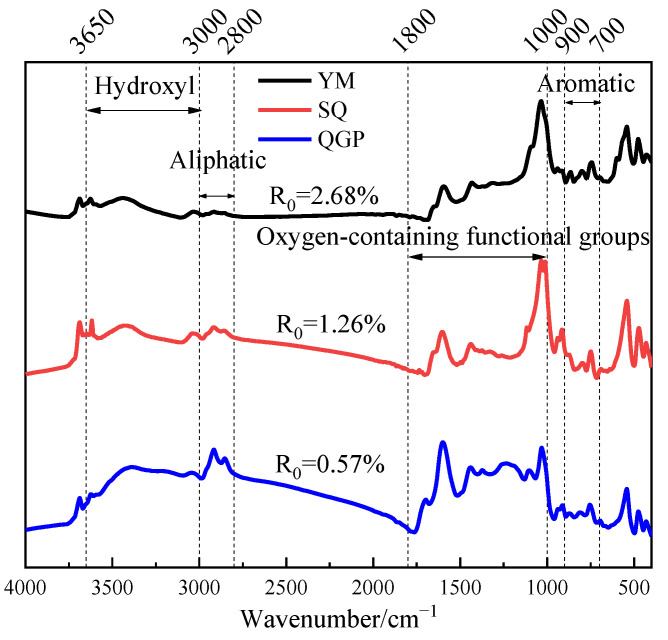
FTIR spectrum of coal sample.

**Table 1 molecules-28-02264-t001:** *DOS* calculation.

Sample	*R*_max_/%	*φ* _750_	*φ* _815_	*φ* _870_	*DOS*
QGP	0.573	60.17%	25.83%	14.00%	0.23
SQ	1.264	67.23%	12.86%	19.91%	0.30
YM	2.678	55.92%	18.62%	25.46%	0.46

Note: *φ*_750_ = A_750_ cm^−1^/[A_750_ cm^−1^ + A_815_ cm^−1^ + A_870_ cm^−1^] *φ*_815_ = A_815_ cm^−1^/[A_750_ cm^−1^ + A_815_ cm^−1^ + A_870_ cm^−1^]. *φ*_870_ = A_870_ cm^−1^/[A_750_ cm^−1^ + A_815_ cm^−1^ + A_870_ cm^−1^] *DOS* = A_870_ cm^−1^/A_750_ cm^−1^.

**Table 2 molecules-28-02264-t002:** Calculation of semi-quantitative structural parameters of anthracite FTIR.

Structure Parameter Name	Structural Parameter Calculation	Absorption Zone
*AR*	CH_ar_ out-of-plane deformation/CH_al_	*A*(900~700)/*A*(3000~2800)
*DOC*	CH_ar_ out-of-plane deformation/C=C	*A*(900~700)/*A*(1600)
*A*(CH_2_)/*A*(CH_3_)	CH_2_/CH_3_	*A*(2900~2940)/*A*(2940~3000)
‘*A*’	CH_al_/(CH_al_+C=C)	*A*(3000~2800)/[*A*(3000~2800) + *A*(1600)]
‘*C*’	C=O/(C=O+C=C)	*A*(1800~1650)/[*A*(1800~1650) + *A*(1600)]
*D*	C=O/C-O	*A*(1800~1650)/A(1330~1030)

Note: *A*(CH_2_)/*A*(CH_3_): length of aliphatic side chain; *AR*: degree of aromaticity; *DOC*: degree of condensation; ‘*A*’: hydrocarbon generation potential; ‘*C*’: maturation level; *A*: fitted peak area.

**Table 3 molecules-28-02264-t003:** Results of semi-quantitative structural parameters.

Sample	*f* _a_	*AR*	*DOC*	*CL*	‘*A*’	‘*C*’	*D*	*R*_max_/%
QGP	0.755	0.44	0.14	6.75	0.24	0.27	2.72	0.573
SQ	0.863	1.13	0.41	4.21	0.26	0.198	0.94	1.264
YM	0.898	3.29	0.56	9.72	0.15	0.15	0.23	2.678

**Table 4 molecules-28-02264-t004:** Industrial Analysis and elemental analysis of coal samples.

Sample	Industrial Analysis, *w* (%)			*R*°_max_	Elemental Analysis, *w* (%)					Atomic Ratio	
	M_ad_ (%)	A_ad_ (%)	V_ad_ (%)		C	H	O *	N	S	H/C	O/C
QGP	2.43	4.06	28.43	0.573	77.75	5.07	14.54	1.43	1.21	0.78	0.14
SQ	0.78	7.85	22.16	1.264	83.08	3.64	9.75	2.34	1.19	0.53	0.088
YM	1.70	13.35	7.44	2.678	85.53	4.52	6.53	1.98	0.44	0.63	0.057

Note: M: Moisture; A: Ash; V: Volatile; ad: Air drying basis; * Oxygen content is calculated by subtraction method;.

## Data Availability

The datasets used and/or analyzed during the current study are available from the corresponding author upon reasonable request.

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
