# Peer review of "Study on the Occurrence Difference of Functional Groups in Coals with Different Metamorphic Degrees"

_molecules, 2023, doi:10.3390/molecules28052264_

Round 1

Reviewer 1 Report

All caption table are missing

Shows the novelty of this study in last paragraph of introduction

What does this research imply?, the final sentence in abstract should be clear

The layout must be adjust especially for the figure

Spacing between number and cm-1 for FTIR results

Author Response

Dear reviewers, Thank you for your useful comments and suggestions on the language and structure of our manuscript. We have modified the manuscript accordingly, and detailed corrections are listed below point by point:

Reviewer 1

All caption table are missing

Shows the novelty of this study in last paragraph of introduction

What does this research imply?, the final senten7ce in abstract should be clear

The layout must be adjust especially for the figure

Spacing between number and cm-1 for FTIR results

I accept the reviewer's opinion and have made modifications in the manuscript. The modifications are marked in blue. The specific modifications are as follows:

1.I add all caption table in front of the tables.

Table 1  Industrial Analysis and elemental analysis of coal samples

Sample

Industrial analysis,w(%)

R°max

Elemental analysis,w(%)

Atomic ratio

Mad(%)

Aad(%)

Vad(%)

C

H

O*

N

S

H/C

O/C

QGP

2.43

4.06

28.43

0.573

77.75

5.07

14.54

1.43

1.21

0.78

0.14

SQ

0.78

7.85

22.16

1.264

83.08

3.64

9.75

2.34

1.19

0.53

0.088

YM

1.70

13.35

7.44

2.678

85.53

4.52

6.53

1.98

0.44

0.63

0.057

Table1:Note: M: Moisture; A: Ash; V: Volatile; ad: Air drying basis; *Oxygen content is calculated by subtraction method

Table 2 DOS calculation

Sample

Rmax/%

φ750

φ815

φ870

DOS

QGP

0.573

60.17%

25.83%

14.00%

0.23

SQ

1.264

67.23%

12.86%

19.91%

0.30

YM

2.678

55.92%

18.62%

25.46%

0.46

Note:φ750=A750 cm-1/[A750 cm-1+A815 cm-1+A870 cm-1]  φ815=A815 cm-1/[A750 cm-1+A815 cm-1+A870 cm-1]

φ870=A870 cm-1/[A750 cm-1+A815 cm-1+A870 cm-1]  DOS=A870 cm-1/A750 cm-1

Table 3 Calculation of semi quantitative structural parameters of anthracite FTIR

Structure parameter name

Structural parameter calculation

absorption zone

AR

CHar out-of-plane deformation /CHal

A(900~700)/A(3000~2800)

DOC

CHar out-of-plane deformation /C=C

A(900~700)/A(1600)

A(CH2)/A(CH3)

CH2/CH3

A(2900~2940)/A(2940~3000)

'A'

CHal/(CHal+C=C)

A(3000~2800)/[A(3000~2800)+A(1600)]

’C’

C=O/(C=O+C=C)

A(1800~1650)/[A(1800~1650)+A(1600)]

D

C=O/C-O

A(1800~1650)/A(1330~1030)

Note: A(CH2)/A(CH3): length of aliphatic side chain; AR: degree of aromaticity; DOC: degree of condensation; 'A': hydrocarbon generation potential; 'C': maturation level; A: fitted peak area;

Table 4 Results of semi quantitative structural parameters

Sample

fa

AR

DOC

CL

'A'

’C’

D

Rmax/%

QGP

0.755

0.44

0.14

6.75

0.24

0.27

2.72

0.573

SQ

0.863

1.13

0.41

4.21

0.26

0.198

0.94

1.264

YM

0.898

3.29

0.56

9.72

0.15

0.15

0.23

2.678

  1. I have Shown the novelty of this study in last paragraph of introduction.

Previous studies have revealed the fugitive states of functional groups in coal and studied the evolution of coal in detail, but the previous studies have studied relatively few changes in the fugitive content of functional groups in different chemical fractions in different coal ranks. Therefore, in this paper, the Fourier infrared characterization technique is used to deconvolute the measured spectra by using Peakfit4.12 data processing software, and the long-flame, coking and anthracite coals of low, medium and high rank coals are characterized, and the functional group contents of each chemical component in the coals are quantified, and the relative contents and trends of each functional group in different coal rank coals are obtained, and the semi-quantitative structural parameters are calculated accordingly. Combining the information of functional group content and semi-quantitative structural parameters, and giving the evolution law of each chemical component in different coal rank, the evolution process of coal rank is discussed, and the structural evolution of coalification in different coal rank and its mechanism are obtained.

  1. I revised the summary and added research imply in final sentence in abstract

n order to quantitatively study the difference of occurrence content of functional groups in coals with different metamorphic degrees, the samples of long flame coal, coking coal and anthracite of three different coal ranks were characterized by FTIR, and the relative content of various functional groups in different coal ranks was obtained. The semi quantitative structural parameters were calculated, and the evolution law of chemical structure of coal body was given. The results show that with the increase of the metamorphic degree, the substitution degree of hydrogen atoms on the benzene ring in the aromatic group increases with the increase of the vitrinite reflectance. With the increase of coal rank, the content of phenolic hydroxyl, carboxyl, carbonyl and other active oxygen-containing groups gradually decreased, and the content of ether bond gradually increased. Methyl content increased rapidly first and then increased slowly, methylene content increased slowly first and then decreased rapidly, and methylene content decreased first and then increased. With the increase of vitrinite reflectance, the OH-π hydrogen bond gradually increases, the content of hydroxyl self association hydrogen bond first increases and then decreases, the oxygen hydrogen bond of hydroxyl ether gradually increases, and the ring hydrogen bond first significantly decreases and then slowly increases. The content of OH-N hydrogen bond is in direct proportion to the content of nitrogen in coal molecules. It can be seen from the semi quantitative structural parameters that with the increase of coal rank, the aromatic carbon ratio fa, aromatic degree AR and condensation degree DOC increase gradually. With the increase of coal rank,A(CH2)/A(CH3) first decreases and then increases, hydrocarbon generation potential 'A' first increases and then decreases, maturity 'C' first decreases rapidly and then decreases slowly, and factor D gradually decreases. This paper is valuable for analyzing the occurrence form of functional groups in different coal ranks and clarifying the evolution process of structure in China.

4.The layout must be adjust especially for the figure

I adjust layout for the figure.

5.Spacing between number and cm-1 for FTIR results

I add the space between number and cm-1.

Reviewer 2 Report

Aromatics may be polycyclic  or graphitic . In that case conclusion becomes difficult .How ever authors intelligently dealt the spectra .Idea is interesting.

It is will be decent if some high-light is focused over it . 

Author Response

I accept the reviewer's opinion and have made modifications in the manuscript.

Reviewer 3 Report

  The manuscript presents a study on functional groups for three different types of coals. These materials were characterized by FTIR spectroscopy and elemental analysis, various functional groups were quantified and semi-quantitative structural parameters were calculated. The authors used data processing software to solve the measured FTIR spectra.

The approach is appropriate, all the materials and methods are completely described, and the results are fully discussed and well explained. The authors have a multitude of bibliographic references.

The manuscript presents interesting data, so I warmly recommend its publication.

Author Response

(The authors gave the same response as above.)
